# Fatal Fulminant Epstein–Barr Virus (EBV) Encephalitis in Immunocompetent 5.5-Year-Old Girl—A Case Report with the Review of Diagnostic and Management Dilemmas

**DOI:** 10.3390/biomedicines12122877

**Published:** 2024-12-18

**Authors:** Magdalena Mierzewska-Schmidt, Anna Piwowarczyk, Krystyna Szymanska, Michal Ciaston, Edyta Podsiadly, Maciej Przybylski, Izabela Pagowska-Klimek

**Affiliations:** 1Department of Pediatric Anesthesiology and Intensive Therapy, Medical University of Warsaw, 02-091 Warsaw, Poland; 2Department of Pediatrics with Clinical Assessment Unit, Medical University of Warsaw, 02-091 Warsaw, Poland; anna.piwowarczyk@wum.edu.pl; 3Department of Pediatric Neurology and Rare Diseases, Medical University of Warsaw, 02-091 Warsaw, Poland; krystyna.szymanska@wum.edu.pl; 4Department of Pediatric Radiology, Medical University of Warsaw, 02-091 Warsaw, Poland; 5Laboratory of Microbiology, University Center of Laboratory Medicine, Medical University of Warsaw, 02-091 Warsaw, Poland; 6Department of Dental Microbiology, Medical University of Warsaw, 02-091 Warsaw, Poland; 7Chair and Department of Medical Microbiology, Medical University of Warsaw, 02-091 Warsaw, Poland; maciej.przybylski@wum.edu.pl

**Keywords:** encephalitis, acute fulminant cerebral edema, children, Epstein–Barr virus (EBV), management algorithm

## Abstract

Introduction: Epstein–Barr virus (EBV) usually causes mild, self-limiting, or asymptomatic infection in children, typically infectious mononucleosis. The severe course is more common in immunocompromised patients. Neurological complications of primary infection, reactivation of the latent infection, or immune-mediated are well-documented. However, few published cases of fatal EBV encephalitis exist. Case presentation We report a case of a 5.5-year-old immunocompetent girl with fulminant EBV encephalitis fulfilling the criteria for the recently proposed subtype Acute Fulminant Cerebral Edema: (AFCE). The child presented with fever, vomiting, altered mental status, and ataxia. Her initial brain CT (computed tomography) scan was normal. On day 2 she developed refractory status epilepticus requiring intubation, ventilation, and sedation for airway protection and seizure control. Magnetic resonance imaging (MRI) scan showed cytotoxic brain edema. Despite intensive treatment, including acyclovir, ceftriaxone, hyperosmotic therapy (3% NaCl), intravenous immunoglobulins (IVIG), corticosteroids, as well as supportive management, on day 5 she developed signs of impending herniation. Intensification of therapy (hyperventilation, deepening sedation, mannitol) was ineffective, and a CT scan demonstrated generalized brain edema with tonsillar herniation. EBV primary infection was confirmed by serology and qPCR in blood samples and post-mortem brain tissue. An autopsy was consistent with the early phase of viral encephalitis. Conclusions This case confirms that normal or non-specific CT and MRI scans do not exclude encephalitis diagnosis if clinical presentation fulfills the diagnostic criteria. The implementation of prophylactic anticonvulsants could improve outcomes. Intracranial pressure (ICP) monitoring should be considered in AFCE for better ICP management. Decompressive craniectomy might be a life-saving option in refractory cases. An encephalitis management algorithm is proposed.

## 1. Introduction

Epstein–Barr virus (EBV), a member of the gamma herpesvirus subgroup of the Herpesviridae family, is probably the most ubiquitous human virus, infecting 90–95% of the population worldwide [1,2]. EBV infects mature epithelial cells of the oropharynx and uterine cervix and selectively invades B-lymphocytes [2,3,4]. It is usually transmitted by oral secretions and less commonly through organ or hematopoietic stem-cell transplantation [5]. The prevalence of infection in children depends on age, geographic location, and socioeconomic status and ranges from 80 to 90% of children aged 2 to 3 years in developing countries to 51% of children by age 4 in Western countries [6]. Improvements in socioeconomic status are associated with declines in EBV antibody prevalence in children [7].

EBV infection is associated with a wide variety of disorders ranging from benign and self-limited illnesses to aggressive malignancies. Primary EBV infection is usually asymptomatic in childhood; in adolescence or adulthood, it induces infectious mononucleosis (IM), a self-limited, usually benign disease with a good prognosis. In some cases, however, it can lead to serious complications involving hematopoietic, cardiovascular, central nervous system (CNS), skin, kidney, and immune system and result in a chronic multisystem disease [8]. After recovery from an acute infection, the virus establishes a long-term latent infection in B-cell lymphocytes. Chronic active EBV infection (CAEBV) is characterized by recurrent or persistent IM-like symptoms lasting for several months and has a poor prognosis, with a mortality rate of up to 43% [9,10]. Chronic or recurrent EBV infection has been closely linked to some immune disorders and cancer [11,12].

Typically, in the Herpesviridae family, EBV is also responsible for neurological damage of varying severity. The most common neurological complication of EBV infection is encephalitis. EBV is the third most common cause of encephalitis within the Herpesviridae family and the seventh most common cause of infectious encephalitis [13,14]. It can lead to serious sequelae: 20% mortality and different types of disabilities in about 20% of patients [15]. EBV infection also contributes to other nervous system diseases not only as a primary neuroinfection but also as an immune-mediated process, including meningitis, myelitis, cerebellitis, polyradiculomyelitis, transverse myelitis, cranial and peripheral neuropathies, and psychiatric abnormalities, Alice in Wonderland syndrome, Guillain–Barre syndrome, Parkinson’s disease, multiple sclerosis, and Alzheimer’s disease [15,16,17].

Neurological complications of EBV infection can result from primary neuroinfection, immune-mediated processes, or reactivation of latent infection, especially in immunosuppressed individuals. EBV invades neurons directly through blood circulation or indirectly via infected B-lymphocytes. It can replicate in the CNS and disrupt the integrity of the blood–brain barrier, causing neuronal damage, inflammation, and demyelination, promoting the degeneration and necrosis of glial cells. Encephalitis can also be the consequence of the immune-mediated process. EBV may share a common antigen with neurological myelin oligodendrocyte glycoprotein, which makes the immune system produce autoimmune T lymphocytes and anti-neuronal antibodies to autoantigens [18,19,20]. The study from China reported one patient developing anti-N-methyl-D-aspartate receptor (anti-NMDA-R) encephalitis during the recovery period from EBV infection [21]. This type of autoimmune encephalitis caused by antibodies against the glutamatergic N-methyl-D-aspartate receptor (NMDA-R) is characterized by five developmental stages, in which psychiatric symptoms predominate in adults and neurological symptoms in pediatric patients.

In children, neurological EBV-associated complications are relatively rare: 0.4–7.5% [9,21]. In a recent study assessing 89 children with EBV infection affecting the nervous system, the authors observed viral encephalitis/meningoencephalitis in 64 cases (71.9%), acute myelitis in two cases (2.2%), ADEM in three cases (3.4%), and GBS in 15 cases (16.9%). Neurological complications in EBV-induced HLH (hemophagocytic lymphohistiocytosis) were observed in four cases (4.5%) and NS-PTLD (nervous system—post-transplant lymphoproliferative disorder) in 1 case (1.1%) [21].

Viral encephalitis/meningitis is the most common neurological complication associated with EBV infection in both adults and children [15,21]. Patients are essentially immunocompetent and young [15]. In children, EBV encephalitis is considered to represent a primary neuroinfection: most children with EBV viral encephalitis have no symptoms of typical EBV infection such as tonsillitis, enlarged lymph nodes, skin rash, and hepatosplenomegaly. The clinical manifestations are not specific: children present with an acute onset of fever, symptoms of intracranial hypertension such as vomiting and headache, altered mental status, convulsions, ataxia, or cranial nerve involvement. Respiratory failure due to brainstem involvement is only rarely observed [21]. The diagnosis of encephalitis resulting from primary EBV infection can be established based on the following: 1. EBV blood serology (the presence of anti-EBV IgM and the absence of anti-EBV IgG confirm primary EBV infection), 2. Detection of EBV DNA in the blood by polymerase chain reaction (PCR) or better quantitative PCR (qPCR) test, 3. Detection of EBV DNA in cerebrospinal fluid by PCR or qPCR (EBV DNA may be absent in CSF if lumbar puncture is performed early in the course of EBV encephalitis, so the second lumbar puncture may be indicated to confirm the diagnosis), 4. qPCR EBV DNA testing from brain biopsy samples, or, as in our patient, post-mortem brain biopsy. Imaging is not specific. CT scans, especially in the early phase, are usually normal [15]. MRI scans are abnormal in 41.5–69% of cases [5,22,23]. The most common locations of MRI lesions were cortical/subcortical (20%), white matter (15.5%), and basal ganglia (11%), followed by the thalamus (9%), brainstem (6.2%), substantia nigra (4.4%), cerebellum (4.4%), and spinal cord (3%). Diffusion restriction and susceptibility changes were observed in 24.4% and 15.5% of cases, respectively. Meningeal involvement was seen in 5–22%. Brain abscesses, subdural effusion/empyema, and hemorrhage are rare [24]. A review carried out by investigators from Lund University suggests that the location of brain damage can have a prognostic value [22].

The management includes symptomatic and supportive treatment, antiviral agents (acyclovir or ganciclovir, less commonly cidofovir and foscarnet), management of raised intracranial pressure, and in selected cases, corticosteroids and/or immunoglobulins.

The overall prognosis of EBV encephalitis in children is relatively good. According to Cheng et al., 68.7% (44/64) of the patients recovered completely, and 14% (9/64) developed disabilities of varying degrees, including mental retardation, motor disorders, language disorders, defecation disorders, and secondary epilepsy. Of the 64 patients, 4 (6.25%) died due to respiratory and circulatory failure caused by brainstem involvement [21]. Therefore, it should be taken into consideration that a severe even fulminant course in EBV encephalitis is possible.

## 2. Case Report

Hereby present a case of fulminant, fatal EBV encephalitis in a 5.5-year-old girl, fulfilling the criteria for a subtype of encephalitis, Acute Fulminant Cerebral Edema (AFCE), as recently defined by Krishnan et al. 2021, [25].

The child was brought to the emergency department (ED) because of altered mental status, problems with verbal contact (inappropriate and missing words), loss of balance (ataxia), and vomiting. The illness began 3 days before with a fever of up to 39 degrees Celsius and abdominal pain lasting 2 days. Her medical history was unremarkable, apart from confirmed influenza 5 weeks earlier. On admission her general condition was good, but neurological examination revealed intention tremor, dysmetria, and a brachybasic gait. The CT scan was obtained (Figure 1A), but no pathology was found. As the child’s condition improved while she was in the ED, she was discharged home at her parents’ request. The next day neurological symptoms recurred, so she was brought back to the ED. Initially, she was conscious, although an atactic gait, tremor, as well as decreased muscle tone were observed. While awaiting admission, she developed trismus and refractory status epilepticus. She was intubated using rapid sequence induction (RSI) for neuroprotection, and a pharmacological coma with thiopental, midazolam, and morphine was initiated to control seizures, alongside 3% NaCl for brain edema prevention. Intravenous acyclovir, ceftriaxone, and vancomycin were started due to suspected encephalitis of unknown etiology. An MRI scan (Figure 2) obtained shortly after initial stabilization revealed cytotoxic brain edema. The lumbar puncture revealed elevated CSF protein (63.7 mg/dL), pleocytosis (21 per μL, 100% lymphocytes), and a positive Pandy test. Unfortunately, testing for EBV in the CSF was not performed, and the lumbar puncture could not be repeated because of the fulminant cerebral edema. A broad spectrum of microbiology, serology, and immunology tests were performed on blood and CSF samples. While awaiting the microbiology results acyclovir and antibiotics were continued, and it was decided to add intravenous (iv) immunoglobulins and corticosteroids.

Initially, inflammatory markers, such as CRP and PCT, were low but later increased during her pediatric intensive care unit (PICU) stay. The results for the antiviral capsid antigen (VCA) VCA IgM and qPCR for EBV were positive (654I IU/mL), while VCA IgG and EBNA IgG (EB anti-nuclear antigen) were negative, confirming a primary EBV infection.

The treatment was continued for another 2 days. The child’s condition seemed stable; however, due to arterial hypotension, noradrenaline and dopamine infusions were started. Her pupils remained equal and reactive to light. On day four, thiopental infusion was reduced, and shortly after, recurrent focal, right-sided seizures recurred. The dosage of sedative drugs was further increased, successfully controlling seizures. Later that evening, without any prodromal signs, her pupils suddenly dilated and became unreactive to light. Recognizing impending herniation, the treatment of the brain edema was immediately intensified: hyperventilation was initiated, and 15% mannitol was added alongside 3% NaCl. On a repeated CT scan (Figure 1B), generalized brain edema, tonsillar herniation, and decreased cortical gray matter attenuation with loss of normal gray-white matter differentiation were evident. Sedation was stopped, but the patient remained unconscious; her pupils remained fixed and dilated. Corneal and gag reflexes were absent, as well as respiratory drive and motor response to painful stimuli. Global hypotonia and lack of deep reflexes were observed. She developed central diabetes insipidus. Primary brainstem areflexia was suspected, and after a period of observation, two series of clinical examinations and cerebral angiography were performed in accordance with national protocol. On day six, the patient was pronounced dead, and life-sustaining treatment was withdrawn.

After her death, the rest of the results were obtained. The antibody testing performed to look for autoimmune encephalitis (onconeural antibodies, anti-MOG, anti-Aqp-4, anti-NMDA, anti-AMPA ½, anti-DPPX, anti-GABAR B1/B2, anti-protein LG1, and CASPR 2) was negative. Additionally, genetic testing, the whole exome sequencing (WES), did not reveal any genetic defect that could predispose our patient to acute encephalopathy triggered by a viral infection. None of the other infective agents tested (over 30 viruses, including tick-borne encephalitis, bacteria, including atypical ones, and fungi) was found. The autopsy result was compatible with an early phase of fatal, fulminant viral encephalitis. EBV etiology was confirmed as EBV was detected in her brain tissue (post-mortem) in 21,190 copies of EBV DNA/0.1 g brain tissue.

## 3. Discussion

We present a patient with an acute, fulminant, and fatal cerebral edema as a consequence of EBV encephalitis in an immunocompetent child with primary EBV infection. This was confirmed by positive IgM VCA EBV antibodies and negative VCA and EBNA IgG antibodies, as well as positive blood PCR for EBV DNA and later confirmed by qPCR in the brain tissue (post-mortem), which detected 21,190 EBV DNA copies/0.1 g of brain tissue.

Acute encephalopathy resulting from encephalitis, complicated by fulminant cerebral edema, is a rapidly evolving and often fatal neurologic condition. This rare phenotype has been previously described in pediatric patients with mortality ranging up to 80% [25,26,27,28,29,30]. No specific pathogen has been identified as the sole cause of this type of encephalitis. However, EBV has rarely been diagnosed as the etiologic factor in such a severe course [21,25,28,29].

Our patient fulfilled the criteria for both the California Encephalitis Project (CEP) for encephalitis and AFCE (acute fulminant cerebral edema) [25]. The CEP criteria include children aged ≥6 months, immunocompetent status, and hospitalization with encephalopathy: depressed or altered level of consciousness lasting 24 h, lethargy, or changes in personality, along with at least one additional finding from the following: fever, seizure, focal neurologic deficit, CSF pleocytosis, electroencephalographic changes consistent with encephalopathy, or neuroimaging suggestive of encephalitis. AFCE represents a recently distinguished subtype of severe encephalitis associated with a rapidly progressing course and high mortality [25]. Its criteria contain fever, altered mental status, and new-onset seizures, followed by progression to diffuse cerebral edema [25]. As fulminant cerebral edema can occur in various conditions such as vascular, neoplastic, infectious, toxic, traumatic, metabolic, or hypoxic-ischemic processes, AFCE is defined as encephalopathy with rapidly progressive elevated intracranial pressure and neurological deterioration, leading to impending or actual cerebral herniation or herniation with brainstem compression in neuroimaging, in the absence of another identified cause of cerebral edema [25].

The pathophysiology of AFCE is not yet well understood. It is likely multifactorial, including hypoxic-ischemic injury, systemic inflammatory response, and genetic predisposition [31]. It is thought that both cytotoxic and vasogenic edema (which may result from systemic inflammatory response) contribute to the development of the condition. These factors may affect the permeability of the blood–brain barrier, which can result in significant cerebral edema and herniation. However, no specific pathogen has been identified so far, but respiratory viruses, including EBV, have been frequently linked to such a course of encephalitis [25]. In the case of EBV infection, a few mechanisms have been proposed. First, damage to neural tissue can be caused by direct lytic infection of neurons, as well as by viral proteins, inflammatory cytokines, and free radicals released from infected endothelial cells of the neurovasculature. Furthermore, CNS-infiltrating B cells and the T cell response against them are likely to cause local bystander damage. The EBV infection may also activate T cells that are cross-reactive with CNS antigens, subsequently inducing an autoimmune response. There is growing evidence that genetic factors can contribute to the development of AFCE due to influenza infection. Thermolabile mutations in carnitine palmitoyl transferase (CPT) II leading to cerebral edema have been reported in Asian children [31]. The CPT system is thought to play a major role in energy metabolism through mitochondrial fatty acid oxidation. As a consequence of these mutations, the use of mitochondrial adenosine triphosphate (ATP) is impaired during periods of high fever, which may lead to brain edema [31]. In SARS-CoV-2 encephalitis, two potential mechanisms of brain damage are postulated: direct invasion of the virus and a severe inflammatory process caused by an extensive release of pro-inflammatory cytokines [31]. Those factors were not suggested in the case of EBV infection, but the disease is much less frequent than influenza or COVID-19. Research into EBV pathology should continue because a better understanding of the underlying mechanisms may lead to the development of better therapeutic strategies.

The neurological condition of our patient severely deteriorated within 24 h, and so did the results of neuroimaging. Interestingly, neither the first CT scan performed on day 1 (Figure 2A) nor MRI performed on day 2 (Figure 1) revealed typical signs of encephalitis. This indicates that the lack of typical changes not only in CT but also in MRI in the early phase does not exclude the diagnosis of viral encephalitis. According to Lan SY et.al. [26], who analyzed the course of acute encephalitis in 1038 children, 298 of whom needed PICU admission, 25 developed acute fulminant cerebral edema (2.4%, 25/1038), our case aligns closely with typical patients with AFCE. She had a febrile prodrome like all (25/25, 100%) of the patients with AFCE, was less than 8 years old like 80% (20/25), had an altered level of consciousness like 72% (18/25) of patients, vomited like 60% (15/25), and had seizures 24–48 h prior to showing signs of fulminant cerebral edema like 76% (19/25) and developed status epilepticus like 48% (12/25) that could be considered refractory like 20% (5/25) of the analyzed patients [26].

The duration from the onset of neurological symptoms to signs of brain herniation in our case was 4 days, compared to the observed in the above study, which ranged from 0 days to 9 days (mean ± SD, 2.7 ± 2.5 days) [26]. On the day of the neurological symptoms’ onset, the girl did not show any signs of cerebral edema on CT. Cytotoxic edema became evident the following day on MRI only after she had developed status epilepticus. Since status epilepticus—particularly if it is refractory to treatment—can be considered a potential risk factor for the development of cerebral edema and increased ICP [26,32], it could be hypothesized that preemptive, prophylactic antiepileptic treatment might be beneficial in cases where acute encephalitis is highly probable. We did not find any studies addressing this issue.

In view of a potentially life-threatening course, we do suggest the need for prompt hospital admission of children with suspected encephalitis to tertiary centers that have access to pediatric ICU and MRI facilities. Furthermore, the diagnosis of encephalitis should not be excluded based on negative imaging findings alone; EBV testing should also be included in immediate diagnostic workups.

As brain edema and/or rhombencephalitis and cerebellitis can make these children prone to brain herniation resulting from lumbar puncture (LP), it is mandatory to respect the contraindications. According to British guidelines [33], brain imaging should precede LP in the presence of any of the following: GCS (Glasgow Coma Scale) < 13 or a decrease in GCS by more than 2 points, focal neurological signs, abnormal posture or posturing, papilledema, seizures until stabilized, bradycardia with hypertension, and doll’s eye movements. If lumbar puncture is deemed unsafe, it should be omitted or delayed until it can be done without the risk of patient harm.

Considering the very poor prognosis of AFCE—for example, as reported by Lan et al. [26], where all 25 patients, including 2/25 with EBV, had bad outcomes (death, persistent vegetative state, or severe neurological sequelae)—we believe that only immediate treatment can potentially save a child’s life. However, the evidence on optimal management strategies is lacking.

The treatment of AFCE includes targeted (e.g., antiviral) and symptomatic and supportive therapy. Acyclovir and ganciclovir effectively inhibit EBV replication in vitro, but their clinical effectiveness is limited, especially in immunocompetent patients, and neither drug is officially approved for this indication [34]. Ganciclovir is considered to be more effective because of better blood–brain barrier penetration. Valaciclovir, an orally administered derivative of acyclovir, has also been used; however, as noted, no antiviral drugs are routinely recommended in EBV encephalitis [30]. Peuchmar et al. suggest that a combination of ganciclovir, intravenous immunoglobulins, and corticosteroids may represent a promising therapeutic trial [15].

Corticosteroids have been used in viral encephalitis as potent anti-inflammatory agents in order to reduce secondary inflammation-mediated damage and/or brain edema. A recent systematic review with meta-analysis did not find data to support corticosteroid use in this setting, although some studies showed positive results [35].

Another therapy used in selected cases of viral encephalitis is intravenous immunoglobulins (IVIg), as they have both anti-inflammatory and immunomodulatory properties. They are relatively safe, and some small studies have demonstrated clinical benefit, but high-quality evidence for their use is lacking [36,37].

In our case, we utilized both corticosteroids and IVIg alongside acyclovir and symptomatic and supportive treatment since the clinical course was extremely severe. These therapies, despite uncertain benefits, appeared to have minimal potential for harm.

Given the absence of proven antiviral agents or other efficacious therapies, increased intracranial pressure (ICP) management is likely the cornerstone of treatment in AFCE. Unfortunately, due to limited evidence, a consensus on optimal ICP management in children, particularly in AFCE, is lacking. In previously reported cases, hyperosmolar therapy, sedation, analgesia, controlled mechanical ventilation, and sometimes hyperventilation were used. There are two options for hyperosmolar therapy. Current pediatric guidelines in different clinical settings recommend hypertonic saline, most commonly 3% solution (2.7–23.4%), as the first-line treatment in the majority of situations [38,39]. Hypertonic saline seems to be more effective than 15–20% mannitol, and conversely, to the latter, it does not induce the risk of a reversed osmotic effect with a destroyed blood–brain barrier, osmotic diuresis, or volume depletion resulting in hypotension [39]. We initially used 3% NaCl in our patient, but as the brain edema was refractory to treatment, 15% mannitol was used as a rescue therapy. Unfortunately, our treatment was not effective—just as in the majority of published AFCE cases [25,26,28,29].

Intubation is commonly required for airway protection and ventilation control, along with pharmacological sedation for brain protection and to control seizures. It is well established that pain and other unpleasant stimuli, as well as cough and straining, increase intracranial pressure. However, it is evident that sedation makes the neurological assessment more difficult or even impossible, so the decision of whether it is needed should be taken individually. Patients with falling levels of consciousness require urgent assessment by a pediatric intensivist for airway protection and ventilatory support, raised intracranial pressure management, and optimization of cerebral perfusion pressure [33]. Currently, it is not known what specific sedatives are preferable. As thiopental commonly induces hypotension, it should be used with caution as second-line therapy in case of refractory increased ICP and/or uncontrollable seizures.

In the case of hypotension, occurring most commonly as an untoward effect of deep sedation, patients with increased ICP need vasoconstrictive and/or inotropic drugs to maintain optimal cerebral perfusion pressure (CPP), as CPP depends on the difference between mean arterial pressure (MAP) and ICP. It should be remembered that in case of severe hemodynamic instability, the diagnosis of neurogenic stunned myocardium or stress-induced (Takotsubo) cardiomyopathy should be taken into consideration [40,41]. These diagnoses seem more probable in encephalitis with fulminant brain edema and/or status epilepticus than in encephalitis without these complications.

Hyperventilation used to be part of the traditional increased ICP management [26,42], but it has been demonstrated to cause vasoconstriction and global cerebral ischemia. Therefore, it should be used only briefly as a rescue measure of impending herniation [39]. Thus, normoventilation is currently recommended, with an optimal target of arterial pCO_2_ of 35–40 mmHg. Other established non-pharmacological interventions include positioning with the elevation of the head from 15 to 30 degrees while maintaining a midline position and temperature control and particularly avoiding fever [39].

Decompressive craniectomy (DC) is the rescue treatment option in refractory brain edema, although its efficacy as well as the timing remain uncertain [33,39,42]. Some reports of successful outcomes in encephalitis-induced intracranial hypertension treated with DC were published [43,44,45]. In the case of our patient, this was not at all considered as the girl progressed to herniation without any prodromal symptoms. Perhaps we could have noticed the impending herniation if we had established ICP monitoring—the modality that in our PICU is used routinely in patients with severe traumatic brain injury.

In fact, since this fatal case, we suggest that in the most severe cases that meet the definition criteria of AFCE, ICP monitoring should be used if available (after this case we had a patient with encephalitis-induced brain managed successfully with ICP monitoring). ICP monitoring allows a prompt reaction to every episode of raised ICP before typical clinical manifestations such as hypertension, bradycardia, or abnormal pupillary reactions might be observed. As we do know that the increased ICP results in poor outcomes in encephalitis [25,26,42], we should aim to detect it early and control it immediately. Imaging is usually not the ideal option, as most PICUs, including ours, do not have bedside CT- or MRI-scanning. If we want to control the brain, the patient must be transported and moved, usually to the supine position. That poses a significant risk with some fatalities; hence, in the case of very unstable patients, we try to limit imaging as much as possible. Considering this, we should look for new bedside methods of ICP monitoring, ideally non-invasive ones. Traditionally the detection of papilledema on fundoscopy has been considered a reliable sign of increased ICP, but it has low sensitivity for intracranial hypertension [46]. Other non-invasive methods have been tried in recent years, including ocular nerve sheath diameter (ONSD) measured using ocular ultrasound. However, evidence is limited; age-based normative values have been proposed [47,48]. Currently, the evidence is scarce and contradictory; therefore, this method cannot be unequivocally recommended [39]. In a recent small study in adults by de Moraes [49], the invasive ICP monitoring was compared to four non-invasive methods: a noninvasive waveform pulse morphology (P2/P1 ratio and Time-To-Peak [TTP) (Brain4care technology), the optic nerve sheath diameter (ONSD), the pulsatility index (PI) by transcranial Doppler (TCD), and a five-item visual scale assessed by brain computed tomography (CT). Those noninvasive methods showed various degrees of correlation with ICP; however, the noninvasive wave morphology monitor and TCD (PI) had the best performances, both isolated and combined [49]. The non-invasive methods, even if none of them is perfect, might help us decide if invasive ICP monitoring should be established and to monitor trends [50]. Another strategy that can be proposed based on our case is bedside electroencephalogram (EEG) monitoring. As suggested by Lan et al. [26], preceding seizures and status epilepticus are significant risk factors for AFCE in children with acute encephalitis. Therefore, effective diagnostic tools to reveal seizures that may be underdiagnosed in sedated patients can have crucial consequences.

We believe the management of AFCE is extremely challenging. Firstly, diagnosis may not be clear, especially in the case of normal CT (just as in our patient), normal or not typical MR brain images, and common delays in specific etiology. The important lesson from our case, (as well as many other AFCE cases cited above), is that imaging, if possible and indicated, should be repeated frequently because it can change rapidly in AFCE. We propose to follow the case definition [25] along with the associated clinical, laboratory, and neuroimaging features of AFCE, with the aim to recognize it early. If the diagnosis of acute encephalitis is highly probable, seizure prophylaxis should be considered, as status epilepticus, even without encephalitis, can lead to brain edema and increased ICP. Lumbar puncture should be performed, sometimes repeated, but it is not always safe in AFCE, and in some cases, it can be contraindicated. Even if we do recognize AFCE, its course may be devastating and uncontrollable because the data on its management is limited and the outcome is usually poor. Acyclovir should always be administered before the diagnosis is established as an agent of low toxicity and high potential for benefit [33,39]. Other therapies have been used, such as intravenous immunoglobulin and intravenous corticosteroids in variable dosages, but no specific treatment was associated with an improved outcome [35,36,37]. Certainly, effective ICP monitoring, with the most reliable still being an invasive method [49], as well as bedside EEG monitoring, potentially might be an important strategy for preventing critical events. However, there are no good-quality studies to support this practice. Research should look for new effective methods of noninvasive ICP monitoring. Future studies should also elucidate the efficacy and timing of decompressive craniectomy (DC) in refractory cases. Because of an extremely high overall mortality in AFCE, perhaps DC should be taken into consideration earlier in the course of the disease than it has been in clinical practice up to now.

Based on our fatal case, we propose the initial management algorithm (Algorithm 1) that could be followed in acute encephalitis. Its applicability needs to be verified in future studies. In the proposed algorithm, we address existing gaps in our knowledge and try to show how they could translate into clinical practices. It is not known how often imaging should be repeated, particularly if the initial scans are normal. Deterioration in GCS > 2–3 points and onset of new neurologic symptoms are recognized as the indications to repeat neuroimaging. Other clinical features have not been defined to date. As we know, seizures in encephalitis are associated with worse outcomes. EEG should be monitored, preferably continuously. In case of suspected or confirmed intracranial hypertension, based on clinical symptoms, and/or imaging, and/or refractory status epilepticus, we propose invasive ICP monitoring to guide brain edema treatment and increased ICP management. We believe that all the steps of the management should be considered, including decompressive craniectomy (DC). It should be remembered that DC can only be beneficial in terms of neurological outcomes, not only lifesaving if the decision is taken before irreversible brain injury occurs. As mentioned above, further studies should address what a timely DC means, including determining when it is neither too early nor too late. Validated scales for neurological assessment of pediatric patients should be developed.

Last, but not least, EBV is a fascinating but highly invasive virus, and still little is known about its pathogenic mechanisms and EBV-mediated immune responses, nor do we have effective antiviral treatments. The administration of rituximab could potentially be an alternative for severe or refractory EBV encephalitis. Recently, two pediatric cases with EBV-related meningoencephalitis, unresponsive to immunoglobulin and corticosteroid therapy, achieved rapid clinical recovery following rituximab administration [51]. Rituximab controls EBV-activated B cell proliferation and eradicates EBV carriage. There is a need to examine the role of rituximab therapy in immunocompetent pediatric patients with EBV encephalitis. This therapy seems promising; however, it can have substantial adverse effects. Therefore, an individual patient’s risk-benefit ratio should be taken into consideration

In conclusion, prompt identification of EBV fulminant encephalitis and early, meticulous management, including all available options, could potentially lead to improved outcomes.
**Algorithm 1.** Suspicion of acute encephalitis—Initial management algorithm.**Clinical suspicion of acute encephalitis—Initial management algorithm**   1. Obligatory hospital admission—fulminant course possible   2. Administer     a. Acyclovir (until other than herpes simplex virus (HSV) causative factor is found or HSV negative on 2nd LP done after 72 h from the onset)     b. Antiepileptic drug(s) if seizures present     c. Hyperosmolar therapy with a first-line bolus of 3% NaCl (3–6 mL/kg) if increased ICP and brain edema confirmed or suspected.     d. III gen cephalosporine (after LP if possible) unless bacterial cause can be ruled out   3. CT/MRI scan to exclude other causes or confirm encephalitis (**of note: normal scan does not exclude encephalitis**)   4. Consider lumbar puncture (LP)—a CT/MRI necessary before LP? Other contraindications to LP? [34]   5. Electroencephalography- EEG   6. **Repeat imaging as clinically indicated and safe for the patient, especially if the first scans are not conclusive.**   7. Consider (potential, though not confirmed benefit)     a. **Prophylactic administration of antiepileptic drugs if diagnosis of encephalitis highly probable even in the absence of seizures**     b. Corticosteroids     c. Iv Immunoglobulins       **In case of GCS (Glasgow Coma Scale) < 8 p., confirmed brain edema and the need for PICU admission act as above, additionally:**   8. Provide:     a. Sedation and analgesia       (some of the sedative drugs have antiepileptic activity but consult neurologist about the use of typical antiepileptic non-sedative drugs like e.g., levetiracetam.)     b. Mechanical ventilation: normoxia, normocapnia- target pCO_2_ 35–40 mmHg   9. Maintain normal or slightly elevated arterial blood pressure having in mind cerebral perfusion pressure (CPP)       **CPP = MAP- ICP**     a. In case of hypotension: start from low dose noradrenaline 0.03 mcg/kg/min titrated to effect.     b. If low dose noradrenaline ineffective perform POC-echocardiography to exclude neurogenic stunned myocardium/stress-related cardiomyopathy.   10. Consider     a. Invasive ICP monitoring     b. Non-invasive ICP monitoring may be useful to observe trends if feasible in your clinical setting.     c. Continuous bedside EEG and sedation monitoring level (BIS, Sedline)     d. In case of refractory intracranial hypertension, seek neurosurgical help: decompressive        craniectomy may be a life-saving option.
Acute alteration in consciousness, cognition, personality or behavior > 24 h **plus any of** (fever/prodromal illness, seizures, focal neurological signs, CSF pleocytosis > 4/μL, CT/MRI compatible with active encephalitis, EEG compatible with active encephalitis (Encephalitis case definition used in ChiMES and ECEPH-UK studies) [51].



## Figures and Tables

**Figure 1 biomedicines-12-02877-f001:**
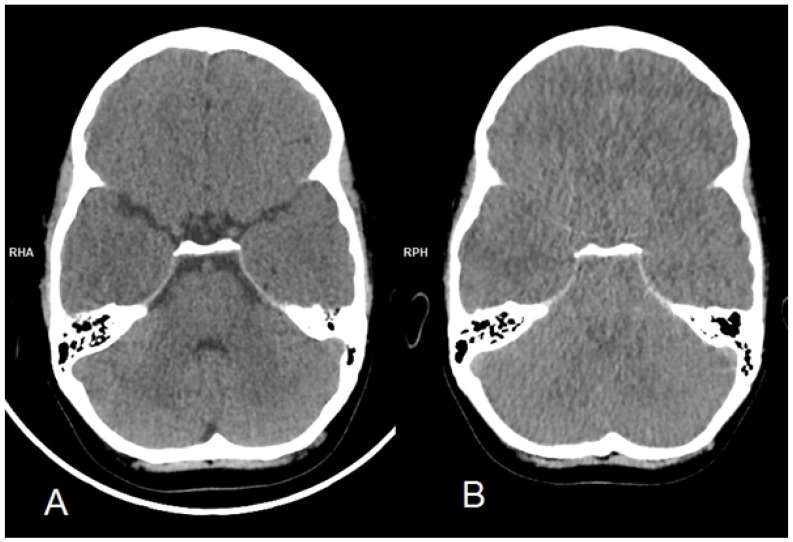
Initial CT scan (**A**) performed at the emergency department on the day of admission shows no abnormalities in the brain parenchyma, with no signs of edema, hemorrhage, or mass effect. Follow-up CT scan (**B**), performed 5 days later due to clinical deterioration and suspicion of tonsillar herniation, reveals diffuse cerebral edema with complete loss of cerebrospinal fluid reserve and evidence of tonsillar herniation through the foramen magnum.

**Figure 2 biomedicines-12-02877-f002:**
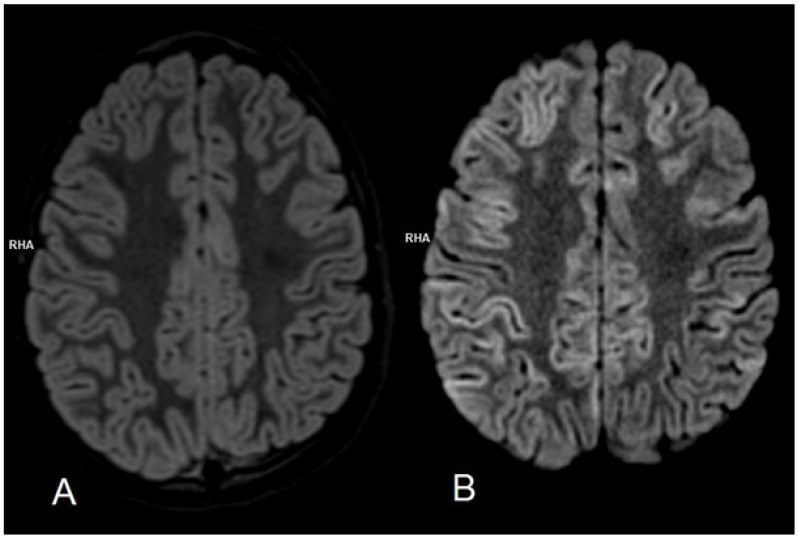
MRI DWI (**B**) sequence reveals diffuse areas of restricted diffusion in the cortical regions of both cerebral hemispheres, consistent with cytotoxic edema. No abnormalities are observed in the T2-FLAIR (**A**) sequence or other performed sequences. These findings suggest a status epilepticus origin rather than an inflammatory process. The pattern is not typical for EBV encephalitis. No areas of abnormal intracranial enhancement were visualized.

## Data Availability

The original contributions presented in this study are included in the article. Further inquiries can be directed to the corresponding author.

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
