# Peer review of "Fatal Fulminant Epstein–Barr Virus (EBV) Encephalitis in Immunocompetent 5.5-Year-Old Girl—A Case Report with the Review of Diagnostic and Management Dilemmas"

_biomedicines, 2024, doi:10.3390/biomedicines12122877_

Round 1

Reviewer 1 Report

Comments and Suggestions for Authors

The authors in their study "Fatal fulminant EBV encephalitis in an immunocompetent 5,5 year-old girl- case report with the review of diagnostic and management dilemmas" reported a case of a 5.5-year-old immunocompetent girl with fulminant EBV encephalitis, and followed up the symptoms and treatment strategies from first day of admission to ICU. The study highlighted the limitations of imaging in early encephalitis diagnosis, the importance of intracranial pressure management, and the potential role of decompressive craniectomy in refractory cases.

I see the study well designed, well presented and well written.

I would ask the authors only to enrich the discussion section of the case report, here are some questions and areas that could be explored further:

  1. Pathophysiology and Diagnostic Challenges:

    • What specific mechanisms might have contributed to the fulminant course of EBV encephalitis in this immunocompetent patient?
    • Could early biomarkers or advanced imaging techniques improve the detection of AFCE before severe symptoms manifest?
  2. Role of Imaging:

    • Considering that early CT and MRI findings were normal, how might imaging protocols be improved to better detect early-stage encephalitis or subtle signs of cytotoxic edema?
  3. Treatment Modalities:

    • Were there any alternative therapies (e.g., plasmapheresis or other experimental approaches) that could have been considered given the refractory nature of the condition?
    • Could the timing of interventions like decompressive craniectomy have altered the outcome?
  4. Seizure Management:

    • Given the refractory status epilepticus, would earlier implementation of prophylactic anticonvulsants have been beneficial?
    • What are the considerations for balancing sedation with the need to monitor neurological function in similar cases?
  5. Intracranial Pressure Monitoring:

    • Should invasive ICP monitoring be standardized in suspected AFCE cases to guide early therapeutic decisions?
  6. Algorithm Development:

    • How might the proposed management algorithm address gaps in current clinical practices?
    • Are there specific scenarios or patient populations where this algorithm could be particularly impactful?
  7. Prognosis and Prevention:

    • What lessons from this case could inform strategies to improve outcomes in similar fulminant cases?
    • Is there a role for prophylactic treatment in children with EBV primary infections who present with neurological symptoms?
  8. Broader Implications:

    • Could this case contribute to identifying risk factors or phenotypic markers for fulminant EBV encephalitis?
    • How does this case align with or challenge the existing literature on severe EBV-related encephalitis in immunocompetent patients?
Comments on the Quality of English Language

The English could be improved to more clearly express the research.

Author Response

Dear Reviewer,

We appreciate the time and effort  you dedicated to providing feedback on our
manuscript. We are certain that your insightful comments will contribute to valuable improvements to our paper, however we have not been able to develop in this paper all the important issues you mentioned. Nevertheless they made us reflect on many issues as well gave us ideas for future research that could hopefully contribute to better encephalitis management. We are incredibly grateful for it. Our answers will be seen below in bold next to your suggestions:

The authors in their study "Fatal fulminant EBV encephalitis in an immunocompetent 5,5 year-old girl- case report with the review of diagnostic and management dilemmas" reported a case of a 5.5-year-old immunocompetent girl with fulminant EBV encephalitis, and followed up the symptoms and treatment strategies from first day of admission to ICU. The study highlighted the limitations of imaging in early encephalitis diagnosis, the importance of intracranial pressure management, and the potential role of decompressive craniectomy in refractory cases.

I see the study well designed, well presented and well written.

Thank you very much for this positive and encouraging opinion.

would ask the authors only to enrich the discussion section of the case report, here are some questions and areas that could be explored further:

  1. Pathophysiology and Diagnostic Challenges:

    • What specific mechanisms might have contributed to the fulminant course of EBV encephalitis in this immunocompetent patient? Thank you for this important remark. We added the whole paragraph on pathophysiology in lines 196-216. Possibly we could speculate that influenza infection a few weeks before the onset of EBV encephalitis might have contribute to it.
    •  
    • Could early biomarkers or advanced imaging techniques improve the detection of AFCE before severe symptoms manifest? Unfortunately we did not find any data on these issues. Certainly future research should go in this direction. Our radiologist and co- author believes all the available MRI sequences were performed. The only hint we found was that in Krishnan study (cited in the text) in 6 acute fatal cerebral oedema (AFCE) patients thalamic changes were detected.
  2. Role of Imaging:

    • Considering that early CT and MRI findings were normal, how might imaging protocols be improved to better detect early-stage encephalitis or subtle signs of cytotoxic edema?
    • As stated above, we did not find studies addressing this important question.
  3. Treatment Modalities:

    • Were there any alternative therapies (e.g., plasmapheresis or other experimental approaches) that could have been considered given the refractory nature of the condition?
    • Because of the fulminant course ( moreover the girl was admitted on  Friday and herniated the following  Monday) we did not have much time to implement experimental therapies. We are not aware of plasmapheresis use in primary encephalitis, and we do not think  it was indicated as the antibody mediated response was unlikely to develop in such a short time. However following your interesting suggestion, we found the first report of Rituximab use in 2 patients with acute EBV encephalitis refractory to IVIG and corticosteroids. So possibly we could have tried this therapy in our patient. We added a paragraph on this topic in lines 356-364.
    • Could the timing of interventions like decompressive craniectomy have altered the outcome? We do believe it could, however because we did not established ICP monitoring we overlooked the threat of impeding herniation.  We discussed this issue in the manuscript. 
  4. Seizure Management:

    • Given the refractory status epilepticus, would earlier implementation of prophylactic anticonvulsants have been beneficial? It needs to be verified in future studies. However, it does seem logical, and toxicity of anticonvulsants is relatively low. Therefore  we propose such approach in our algorithm, as risk-benefit ratio in severe cases seems to be on the side of benefit to the patient. 
    • What are the considerations for balancing sedation with the need to monitor neurological function in similar cases?
    • As we discussed in the text there are some definite indications for intubation, ventilation and sedation eg.  decreasing level of consciousness, brain edema and intracranial  hypertension, and/or refractory status epilepticus. There are some clinical scenario when clinical decisions can be challenging. In case of agitation or aggression, sedation should be used with caution.
  5. Intracranial Pressure Monitoring:

    • Should invasive ICP monitoring be standardized in suspected AFCE cases to guide early therapeutic decisions? We propose such an approach, therefore we  suggested to consider ICP-monitoring in suspected AFCE cases in the encephalitis management algorithm. We believe that because this method enables continuous control of ICP, allowing timely therapeutic interventions (as well as we have positive experience with patients with traumatic brain injury), AFCE cases should benefit from it.   There are a few published case reports mentioned in our paper of AFCE cases with good outcome, possibly due to decompressive craniotomy.  However there is no evidence-based data to support it and more studies are needed to be able to give specific recommendations - both to the specific indications and to its timing. 
  6. Algorithm Development:

    • How might the proposed management algorithm address gaps in current clinical practices? 
    • Thank you for this question. We added a paragraph that underlines the gaps and proposes how to address them in the clinical management lines 343-355. 
    • Are there specific scenarios or patient populations where this algorithm could be particularly impactful? We are not able to answer this valid question based on available data. Definitely patients with high risk for severe course, but as mentioned above, unfortunately  we do not have reliable early markers.
  7. Prognosis and Prevention:

    • What lessons from this case could inform strategies to improve outcomes in similar fulminant cases?
    • The lessons we learnt: normal/ not-typical CT nor MR scan do not exclude the encephalitis diagnosis. They should be repeated if clinically indicated. Prophylactic anticonvulsants should be considered and optimal seizure detection and control is mandatory. ICP monitoring is a useful tool to guide ICP management. In case of refractory hypertension decompressive craniotomy should be considered. We propose them in the algorithm.
    • Is there a role for prophylactic treatment in children with EBV primary infections who present with neurological symptoms? We think that optimal antiepileptic therapy, ICP management, iguided by ICP monitoring n severe cases, and generally supportive treatment is important. Acyclovir, considering its low toxicity should be always administered, until it is 100% sure that it is not needed.
  8. Broader Implications:

    • Could this case contribute to identifying risk factors or phenotypic markers for fulminant EBV encephalitis?  The only thing we could think of was the influenza infection a few weeks before. It is known that viral infections, particularly VZV but also influenza can induce a state of immunosuppression. This, however, do not explain why, the EBV infection was fatal in our patient.
    • How does this case align with or challenge the existing literature on severe EBV-related encephalitis in immunocompetent patients? The fulminant course of EBV encephalitis is rare in immunocompetent children. We discussed the existing data in the manuscript. Thank you again for your inspiring questions. We also asked the native speaker to help us improve the language of the manuscript. Magdalena Mierzewska-Schmidt on behalf of the authors.

Reviewer 2 Report

Comments and Suggestions for Authors

Epstein-Barr virus (EBV) is a herpes virus that causes human infections and has the common characteristic of demonstrating a latency state (with the possibility of reactivation) after the first infection of the host. EBV is responsible for mononucleosis infection, with contagion prevalently through saliva and respiratory secretions. The severe course is more common in immunocompromised patients, while in immunocompetent patients the course is usually favorable. In this case report, the authors describe the case of a 5.5-year-old immunocompetent girl with fulminant encephalitis from EBV and propose an algorithm for the management of encephalitis.

The topic covered falls within the aims and scope of the Special Issue and can certainly be of interest to researchers in the field.

Overall, the structure of the manuscript is well done and reports interesting data.

However, some changes are requested, as reported below.

In the keywords: Please replace the acronym EBV with the full name.

For clarity, please add a sentence before the sentence lines 79-81 (In the 79 study from China…). For example: Encephalitis caused by antibodies against the glutamatergic N-methyl-D-aspartate receptor (NMDA-R) is a recently described disease, characterized by five developmental stages, in which psychiatric symptoms predominate in adults and neurological symptoms in pediatric patients.

Check the entire text and add the full name before the acronym the first time it is mentioned in the text. For example: Computed Tomography (CT) of the brain (line 23).

As requested above: Magnetic Resonance Imaging, MRI (line 25).

Lines 97-100. Please check the entire text to clarify some sentences. For example, the sentence reported in lines 97-100.

Line 261. Check typos and provide more details in the caption of Figure 1.

Add a list of abbreviations used in the text.

Author Response

Dear Reviewer,

We would like to thank you very much for your positive review as well as insightful remarks that will greatly contribute to a better and clearer content of the final manuscript.

Below we reply to your comments in bald:

Epstein-Barr virus (EBV) is a herpes virus that causes human infections and has the common characteristic of demonstrating a latency state (with the possibility of reactivation) after the first infection of the host. EBV is responsible for mononucleosis infection, with contagion prevalently through saliva and respiratory secretions. The severe course is more common in immunocompromised patients, while in immunocompetent patients the course is usually favorable. In this case report, the authors describe the case of a 5.5-year-old immunocompetent girl with fulminant encephalitis from EBV and propose an algorithm for the management of encephalitis.

The topic covered falls within the aims and scope of the Special Issue and can certainly be of interest to researchers in the field.

Overall, the structure of the manuscript is well done and reports interesting data.

We greatly appreciate your positive opinion.

However, some changes are requested, as reported below.

  1. In the keywords: Please replace the acronym EBV with the full name. We have done it.
  2. For clarity, please add a sentence before the sentence lines 79-81 (In the 79 study from China…). For example: Encephalitis caused by antibodies against the glutamatergic N-methyl-D-aspartate receptor (NMDA-R) is a recently described disease, characterized by five developmental stages, in which psychiatric symptoms predominate in adults and neurological symptoms in pediatric patients. We added what you requested (lines 77-80).
  3. Check the entire text and add the full name before the acronym the first time it is mentioned in the text. For example: Computed Tomography (CT) of the brain (line 23) and As requested above: Magnetic Resonance Imaging, MRI (line 25).We have done it. 
  4. Lines 97-100. Please check the entire text to clarify some sentences. We did as you advised. For example, the sentence reported in lines 97-100. Thank you very much, actually it was not very clear, we tried to rewrite it and hope it is more understandable now. The new text is in lines 94-100.
  5. Line 261. Check typos and provide more details in the caption of Figure 1. We corrected the typos in line 261. We added a sentence to the caption of a Figure 1.  To your knowledge, the following sequences were used: T2 tra, T2 dark-fluid 3D, T1 mprage 3D pre- and post contrast, DWI, SWI.  Our co-author who is a radiologist believes that we did everything that was available, however the results were not typical for viral encephalitis. Unfortunately, MRI could not be repeated due to the patient's condition. 
  6. Add a list of abbreviations used in the text. We did it as you kindly advised: lines 370-400.Thank you again,  we greatly appreciate the time and effort that you dedicated to providing the feedback to our manuscript, Best regards, Magdalena Mierzewska-Schmidt on behalf of the authors.